# Dynamic Response Analysis of Offshore Converter Station Based on Vector Form Intrinsic Finite Element (VFIFE) Method

**Zhenzhou Sun [1,2], Yang Yu [3], Huakun Wang [4,\*], Shanshan Huang [1,2] and Jiefeng Chen [1,2]**

[1] Key Laboratory of Far-Shore Wind Power Technology of Zhejiang Province, Hangzhou 311122, China;
sun_zz@hdec.com (Z.S.); huang_ss3@hdec.com (S.H.); chen_jf5@hdec.com (J.C.)
[2] Power China Huadong Engineering Corporation Limited, Hangzhou 311122, China
[3] State Key Laboratory of Hydraulic Engineering Simulation and Safety, Tianjin University,
Tianjin 300072, China; yang.yu@tju.edu.cn
[4] Fujian Key Laboratory of Digital Simulations for Coastal Civil Engineering, Department of Civil Engineering,
Xiamen University, Xiamen 361005, China
\* Correspondence: tjuwhk@tju.edu.cn

**Abstract:** This study aims at proposing a simulation method for an offshore converter station platform (OCS) under dynamic loading. A user-defined in-house FORTRAN code was developed based on the Vector Form Intrinsic Finite Element (VFIFE) method, and the numerical model was validated by test data. After model validation, the dynamic behavior of the OCS was carefully studied and the effect of different loading conditions, including seismic, hydrodynamic and wind load, on the dynamic behavior of OCS was investigated. The time history of the structural response was obtained, and the relationship between the structural peak response and the peak value of the seismic load was also shown. It indicated that water damping accelerates energy consumption, while the effect of hydrodynamic and wind load has little influence on the cases studied in this work. Generally, the peak response increases almost linearly with the increase in the peak acceleration of the ground, and the seismic propagating direction has a great impact. In addition, both a whipping effect and stretching-squeezing effect were observed, and the vertical acceleration response of the valve hall deck is much higher than other structures, which is caused by the relatively lower local rigidity of the large-span structure and the inertia force caused by the valve tower.

**Keywords:** VFIFE; seismic; offshore converter station platform

## 1. Introduction

In recent years, marine wind power has been greatly developed. To improve the efficiency of electricity transmission, the first choice is to adopt a large-scale electrical platform. Due to a large volume of electrical equipment arranged on the upper deck, the offshore converter station platform (OCS) is characterized as a top-heavy structure. The OCS needs to resist the harsh marine environmental loads, especially under the combined action of wind, wave, current and seabed erosion. In addition, it is also threatened by sudden extreme loads, such as sea ice, typhoons and earthquakes [1]. In these cases, the inertial force and the base bending moment may be considerable, the extreme loads will increase the vibration of the structure, and may lead to the collapse of the platform. The choice of the platform type is closely related to the geological condition, the marine hydrological parameters and the structure's weight [2]. Generally, a gravity foundation is suitable for shallow water and good geological conditions; a single-pile foundation is commonly used when the total weight of the superstructure is less than 1000 t, and the jacket foundation is the first choice when the superstructure is heavier (>1000 t). As the total weight of the superstructure of OCS is large, a jacket foundation is commonly used.

For offshore platforms, a strong earthquake may lead to serious damage. Studies on the dynamic behavior of offshore structures that have experienced earthquakes are necessary [3–5]. In recent decades, both theoretical and experimental studies on the structural

response of fixed offshore platforms under seismic load have been carried out by many researchers [3–14].

The seismic response of the traditional steel jacket platform under strong earthquakes has been analyzed by Bea et al. [3], who indicated that the seismic design criterion proposed by American Petroleum Institute (API) specifications for the offshore platform structure is reliable in view of the anti-seismic performance. Chandrasekaran et al. [4] analyzed the dynamic behavior of a triangular tension leg platform (TLP) under the combination of high sea waves and bidirectional (horizontal and vertical) seismic excitation. The seismic forces make tether tension unbalanced, which leads to a nonlinear variation in the tether tension. More recently, the effect of five different hysteresis models on the dynamic behavior of Moment Resisting Frame (MRF) structures under seismic load was investigated by Huang et al. [5], and it indicated that the strength degradation reduces the overall drift limit of the structure significantly, while the effect of stiffness degradation is marginal. The dynamic response of jack-up platforms under the effects of wave, wind, earthquakes and tsunami forces was investigated by Zaid et al. [13] by the finite element method, and the simulation results proved that the jack-up platform hull experienced maximum deformation under high earthquake intensity and with the direction of tsunami waves applied at 45° (east–west). In addition, there are a number of studies focusing on the dynamic behavior of bottom-fixed wind turbines under seismic load [1,2,7–14], where both numerical [1,8,10] and experimental studies [8–10] have been carried out. The effect of loading combinations [8,10], water levels and waveforms [9], and geometric imperfections [11] have been carefully studied. A representative work is that given by Wang et al. [10], where both experimental and numerical analyses of multiple offshore wind turbines under seismic, wind, wave, and current loads are carried out.

Studies on the earthquake suppression measures have also been undertaken by some researchers [15–17], such as the cylindrical [15] and spherical tuned liquid damper [16], where scale test models were designed to investigate their performance. It indicated that the ratio of the fundamental sloshing frequency of liquid to the natural frequency of the platform is the key factor to control earthquake response, and a larger water-mass to platform-mass ratio is also useful for vibration reduction [15].

Although much research on the dynamic characteristics of the jacket platform and wind turbine has been carried out, research on the dynamic characteristics of OCS characterized by a top-heavy structure under the earthquake load has been rarely reported. In the authors' previous work, both experimental [2] and numerical analysis [1] (commercial software ABAQUS AQUA was used) of OCS have been reported. As a further extension of previous work, this study aims to propose a new numerical simulation method for dynamic analysis of OCS. A user-defined in-house FORTRAN code based on VFIFE was developed to realize the simulation. The VFIFE method was first proposed by Shih [18] and Ting [19,20]. The concept of path unit used in the VFIFE enables the calculation of stress of the element by engineering stress and micro-strain when the problems of large displacement, collision and fracture are analyzed. More importantly, it is more convenient to use the VFIFE method to add or subtract elements and change boundary conditions. The discontinuous behavior of the structure can be dealt with simply, while the traditional finite element method does not have the concept of a path unit. The VFIFE method has attracted the attention of many researchers; it has been applied in the dynamic analysis of a marine riser under hydrodynamics [21,22] and the vibration and collapse of two-dimensional cable-stayed bridges under earthquakes [23], while the application of the VFIFE method in the seismic characteristic analysis of OCS has not been reported. In view of this fact, an in-house FORTRAN code based on the VFIFE method was developed and calibrated, which lays a solid foundation for the subsequent highly nonlinear pushover and collapse analysis of OCS under extreme load.

## 2. Numerical Model and Simulation Scheme

### 2.1. Governing Equation

Based on the basic principle of the VFIFE, the OCS was discretized into a series of mass nodes connected by a three-dimensional beam element (see Figure 1). The motion of the mass point was decomposed into path units in the time domain; the shape and position of the OCS were described by tracing the nodal movements based on Newton's Second Law. The corresponding governing equation (for small vibrations) can be written as follows [21,24]:

$$\mathbf{M}_a \frac{d^2\mathbf{d}_a}{dt^2} + \alpha \mathbf{M}_a \frac{d\mathbf{d}_a}{dt} = \mathbf{P}_a(t) + \mathbf{f}_a(t) \quad (a = 1, 2, 3, \dots, N) \tag{1}$$

$$\mathbf{I}_a \frac{d^2\boldsymbol{\theta}_a}{dt^2} + \alpha \mathbf{I}_a \frac{d\boldsymbol{\theta}_a}{dt} = \mathbf{Q}_a(t) + \mathbf{m}_a(t) \quad (a = 1, 2, 3, \dots, N) \tag{2}$$

$$\mathbf{M}_a = \mathbf{diag}(M_a, M_a, M_a), \quad \mathbf{I}_a = \begin{bmatrix} I_{xx} & I_{xy} & I_{xz} \\ I_{yx} & I_{yy} & I_{yz} \\ I_{zx} & I_{zy} & I_{zz} \end{bmatrix} \tag{3}$$

$$\mathbf{d}_a = \{x_a, y_a, z_a\}^T, \quad \boldsymbol{\theta}_a = \{\theta_{ax}, \theta_{ay}, \theta_{az}\}^T \tag{4}$$

$$\mathbf{P}_a = \{P_{ax}, P_{ay}, P_{az}\}^T, \quad \mathbf{Q}_a = \{Q_{ax}, Q_{ay}, Q_{az}\}^T, \quad \mathbf{f}_a = \{f_{ax}, f_{ay}, f_{az}\}^T, \quad \mathbf{m}_a = \{m_{ax}, m_{ay}, m_{az}\}^T \tag{5}$$

where $\mathbf{M}_a$ is the mass of the point, subscript $a$ is the node number; $\mathbf{d}_a$ is the position vector; $t$ is time; $\alpha$ is the damping parameter; $\mathbf{P}_a(t)$ and $\mathbf{f}_a(t)$ are the external and internal force, respectively; $N$ is the number of points; $\mathbf{I}_a$ is the matrix of the moment of inertia of point; $\boldsymbol{\theta}_a$ is the angle vector; and $\mathbf{Q}_a(t)$ and $\mathbf{m}_a(t)$ are the external and internal moment, respectively.

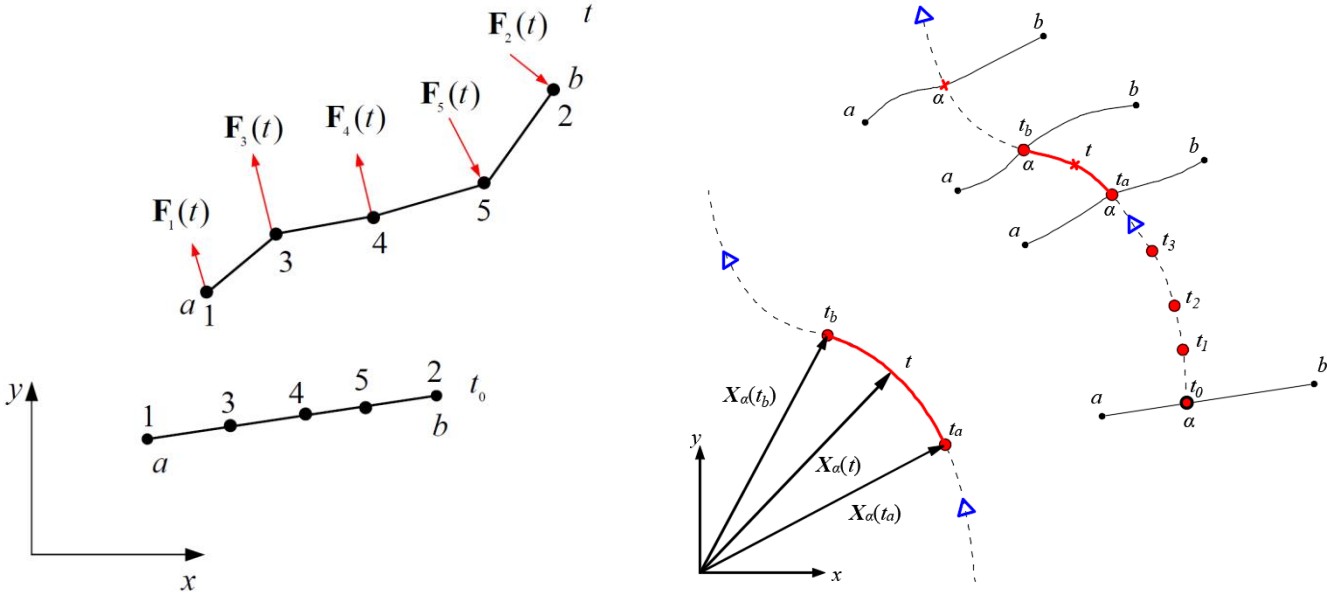

(**a**) Point value description of a beam element        (**b**) Path unit

**Figure 1.** Schematic of the VFIFE method.

As the stiffness damping of the jacket platform structure is far less than the mass damping, many researchers have pointed out that the stiffness damping has little influence on the time-history response of the civil structure [25]. In addition, the overall stiffness matrix is not formed in the calculation process of the VFIFE method; thus, stiffness damping

is not considered in this work. The damping coefficient can be calculated based on the Rayleigh damping coefficient

$$\left[\alpha_1^i\right] = \frac{4\xi}{\omega_{\mathrm{m}} + \omega_{\mathrm{n}}}[\omega_{\mathrm{m}}\omega_{\mathrm{n}}] \tag{6}$$

where $\xi$ is the equivalent damping ratio of structure, which has been determined in the authors' previous work ($\xi$ = 5.73% and 8.68% in the case of without and with water) [2], and $\omega_{\mathrm{n}}$ is the circular frequency corresponding to the $n$th mode. In this work, the calculated natural frequencies of OCS under dry and wet condition are $\omega_1$ = 5.09, $\omega_2$ = 5.40 and $\omega_1$ = 5.01, $\omega_2$ = 5.31, respectively. According to the eigenvalue extraction analysis, the corresponding damping coefficients are 0.601 and 0.908, respectively, and then Equations (1) and (2) are time-integrated using the central difference method; more details can be found in the authors' previous work [21,26,27].

### 2.2. External Force

The external forces acting on the structure include the concentrated loads (moment) acting on the node directly and the equivalent nodal force (moment) of the element distribution load. More details on the derivation of the equivalent nodal force (moment) can be found in the work of Ding [24]. The distributed forces acting on a submerged leg of the OCS include the gravity, buoyancy and hydrodynamic loads, while the seismic load is treated as a concentrated load acting on the base node of the legs. Some details of the force considered in this work are shown in the following sections.

### 2.2.1. Gravity and Buoyancy

The gravity $\mathbf{p}_g$ and buoyancy $\mathbf{p}_b$ of every unit length of element can be written as follows:

$$\mathbf{p}_g = \left(\rho A_s + \rho_I \frac{1}{4}\pi D_I^2\right)\mathbf{g} \tag{7}$$

$$\mathbf{p}_b = -\rho_f \frac{1}{4}\pi D_o^2 \mathbf{g} \tag{8}$$

where $\rho$ is the density of the pipe, $A_s$ is the sectional area of the pipe, $\mathbf{g}$ is the gravitational acceleration, $\rho_I$ is the density of the material inside the pipe, if there is no filler, $\rho_I = 0$; $D_I$ is the inner diameter of the pipe; $\rho_f$ is the density of water; and $D_o$ is the outside diameter of the pipe.

### 2.2.2. Hydrodynamic Load

As performed by other researchers [21,22], the Morison equation is also used to calculate the wave and current load. Note that only the transverse (perpendicular to the axis of the cylinder) drag force was considered, while the tangential (along the axial direction of piles) drag force was ignored. The transverse drag forces acting on the unit length of the cylinder can be written as [28]

$$\mathbf{f} = \frac{1}{2}C_D\rho_f D_o \mathbf{U}_n|\mathbf{U}_n| + C_M\rho_f \frac{\pi D_o^2}{4}\dot{\mathbf{U}}_n \tag{9}$$

where $C_D$ is the drag force coefficient perpendicular to the axis of the cylinder; $\rho_f$ is the density of water; $D_o$ is the outside diameter of the cylinder; $C_M$ is the inertia force coefficient ($C_M = C_a + 1$); $\mathbf{U}_n$ and $\dot{\mathbf{U}}_n$ are the velocity vector and the acceleration vector of the water point that perpendicular to the cylinder axis, respectively. The value of $C_D$ and $C_M$ depends on the shape and surface roughness of the pile; for a cylindrical steel pile with a smooth surface, the values of $C_D$ and $C_M$ are 0.65 and 1.6, respectively [28], which was used in this work. The value of $\mathbf{U}_n$ and $\dot{\mathbf{U}}_n$ are determined by wave theory (Airy wave or Stokes wave theory was used based on the wave parameters). If the influence of current velocity should

be considered, the current velocity ($\mathbf{U}_{f,n}$) should be added on the basis of the water point velocity ($\mathbf{U}_{w,n}$), which yields

$$\mathbf{U}_n = \mathbf{U}_{f,n} + \mathbf{U}_{w,n} \tag{10}$$

Shear flow is assumed below the mean water surface and steady flow is assumed above the mean water level, as shown in Figure 2 [29,30].

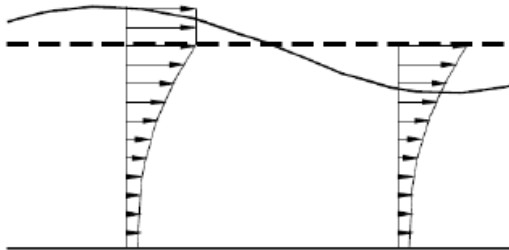

**Figure 2.** Schematic of the current profile.

### 2.2.3. Wind Load

For the calculation of the wind load, $F$, an average wind pressure ($p$) was applied, which is

$$F = pA \sin \beta$$
$$p = \mu_s \mu_z p_0, \quad p_0 = \tfrac{1}{2}\rho_a u_0^2 \tag{11}$$

where $A$ is the projected area of the wind area perpendicular to the force; $\beta$ is the angle between wind direction and wind area; $\mu_s$ is the shape factor of wind load, which is related to the building shape, for cylinder $\mu_s = 0.5$, for a plane $\mu_s = 1.5$; $\mu_z$ is the variation coefficient of the wind pressure along the height; $p_0$ is the basic wind pressure, which can be calculated by Bernoulli equation showing in Equation (11); $\rho_a$ is the density of air, taken as 1.205 kg/m$^3$; and $u_0$ is the basic wind speed. The wind load is regarded as a uniform force acting on the beam, note that the wind load only acts on the beam above the free water line, and the beams within the shelter area are not affected.

### 2.2.4. Seismic Load

In this study, the seismic load is applied by an acceleration boundary condition, as the structure is equivalent to a set of points, the seismic action is equivalent to an external force acting on the base nodes of the jack-up legs, and the equivalent seismic load is

$$F^{eq} = -KM\frac{d^2 x_g}{dt^2} \tag{12}$$

where $F^{eq}$ is the equivalent seismic load acting on the base nodes; $K$ is the ratio of seismic acceleration to gravitational acceleration, $K = a/g$, and $\frac{d^2 x_g}{dt^2}$ is the earthquake acceleration.

### 2.3. Internal Force

The internal force acting on node $a$ is the resultant force from the elements sharing this node:

$$\mathbf{f}_a(t) = \sum_{i=1}^{N_e} \mathbf{f}_a^i, \quad \mathbf{m}_a(t) = \sum_{i=1}^{N_e} \mathbf{m}_a^i \tag{13}$$

where $N_e$ is the number of elements connected to the node $a$, $\mathbf{f}_a^i$ is the internal force of the $i$-th element acting on the node and is the internal moment.

## 3. Model Validation

Based on the basic theory mentioned in Section 2, a user-defined in-house FORTRAN code (called VFOSP) was developed, and a numerical model consistent with the experimental study given by the authors previously [1,2] was established, detailed information is shown in Figures 3 and 4. Some specific monitored points are also marked, where the

displacement or acceleration during the test was recorded. The material used in the model test is plexiglass, and the basic information is shown in Table 1 [1,2]. Only the elastic analysis was performed in this work. The dynamic behavior of the electrical platform structure suffered an artificial seismic wave generated by the API specification spectrum (denoted as YS−A−0.25 g, where YS indicates the test are conducted in water and bidirectional seismic loads are applied (along the $X-$axis and $Z-$direction); A means the API wave, and 0.25 g is the peak value of the seismic load) was investigated, which was also used to verify the accuracy of numerical methods. Detailed information of the seismic load is shown in Figure 5. The water depth used in the test was 0.32 m, and no wave or current load was applied.

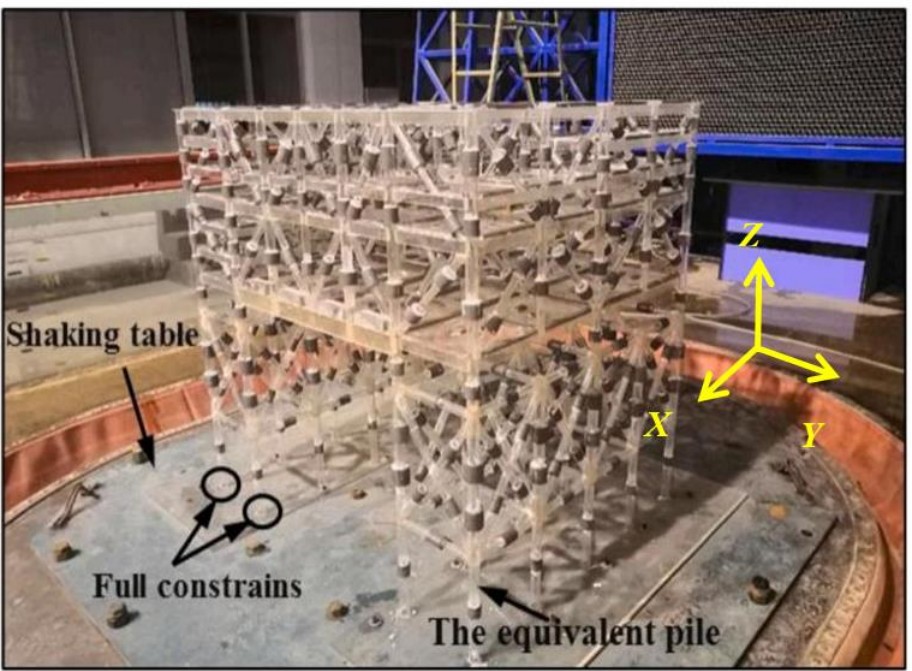

**Figure 3.** Physical model test layout of the electrical platform [1,2].

**Table 1.** Material and model parameters of the numerical model [2].

| Material | $E$ (GPa) | Dynamic Elasticity Modulus, $E_D$ (GPa) | Poisson's Ratio, $v$ | Density, $\rho_s$ (kg/m³) | Water Depth, $h$ (m) |
|---|---|---|---|---|---|
| Plexiglass | 2.62 | 3.91 | 0.42 | 1201.2 | 0.32 |

### 3.1. Convergence Study of the Incremental Size

The central difference method is used during the time-integration, which is conditionally stable. The incremental size will affect the computational efficiency and the accuracy of the simulation results. A too-small increment size results in a significant increase in calculation time, and the output database is too large, which is not convenient for post-processing. While the critical information could be lost if a too-large increment size is used, and the calculation may not converge. To find an appropriate incremental size, where both accuracy and computational efficiency can be satisfied, a convergence study of the incremental size was carried out, and the response of OCS under unidirectional seismic loads (along the $X-$axis) is shown in Figure 6. It indicates that the calculation converges as the increment size is smaller than $1 \times 10^{-5}$. A material point suffered the internal force of the structure

can be simplified as a particle-spring system with a single degree of freedom; the natural angular frequency of this system can be written as follows

$$\omega = \sqrt{\frac{k}{m}} \tag{14}$$

where $k$ is the spring stiffness, and $m$ is the mass of the point. When the explicit central difference method is used to solve the particle's motion equation, the critical time increment size can be determined as follows

$$\Delta t_c = 2\sqrt{\frac{m}{k}} \tag{15}$$

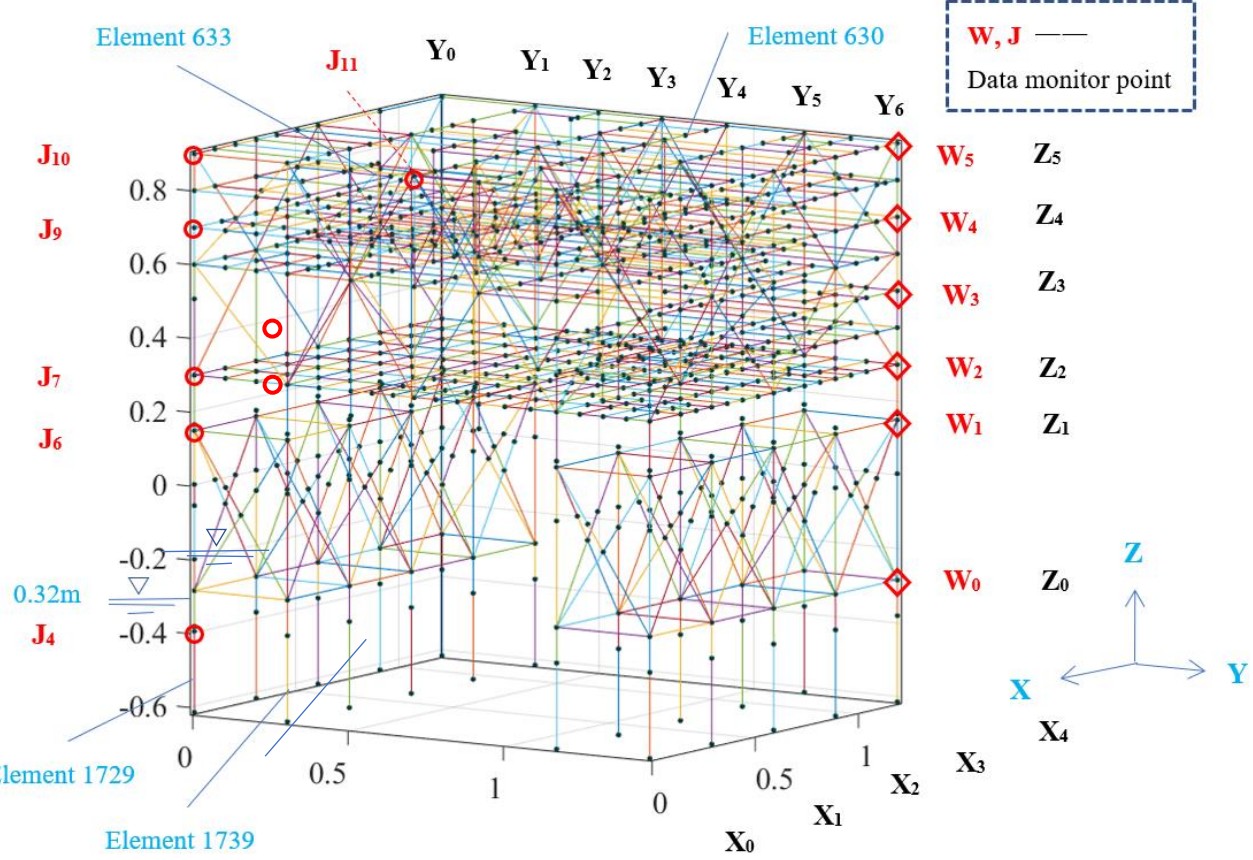

**Figure 4.** Numerical model of the electrical platform structure (Unit: m, $X_i$ (i = 0, 1, ... , 4), $Y_i$ (i = 0, 1, ... , 6) and $Z_i$ (i = 0, 1, 2, 3, 4, 5) show the location of different cross section).

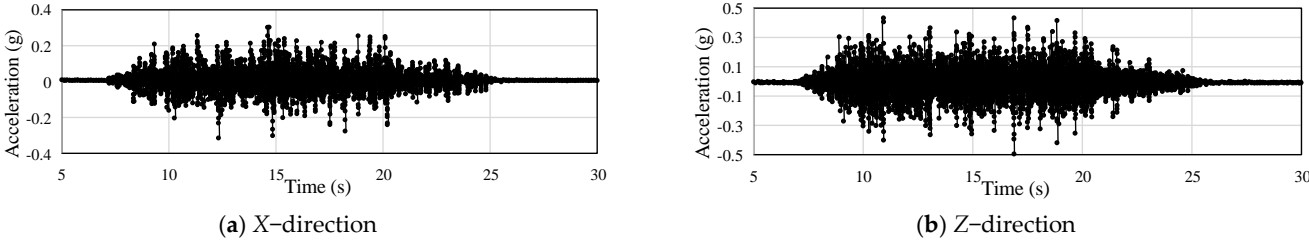

(**a**) $X$−direction

(**b**) $Z$−direction

**Figure 5.** The seismic load used in the test; the artificial seismic wave generated by the API specification spectrum.

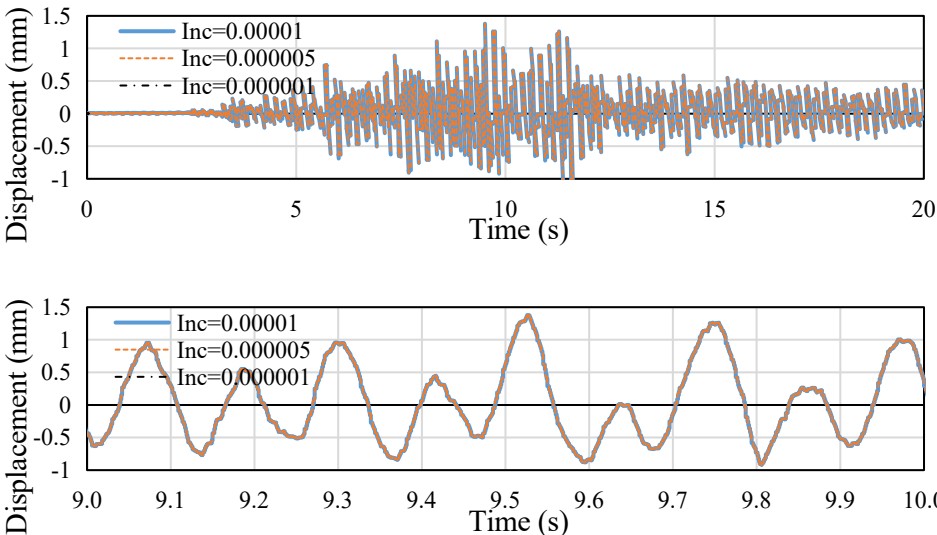

**Figure 6.** Comparison of simulation results of different incremental sizes in air condition (denoted as WD−A−0.25 g).

The internal force between the material point can be divided into axial tensile, shear and bending effect, where the axial tensile stiffness is maximum and the bending stiffness is smallest. Thus, the axial tensile stiffness $k_t$ can be used to estimate the lower limit of the critical increment size, approximately, which yields

$$\Delta t_c = 2\sqrt{\frac{m}{k_t}} = 2L\sqrt{\frac{\rho}{E}}$$
$$k_t = \frac{EA}{L}$$

(16)

where $E$ is the elastic modulus, $A$ is the sectional area of the component, $L$ is the characteristic length of the beam element and $\rho$ is the mass density of the target material. According to Equation (16), the theoretical value of $\Delta t_c$ can be obtained based on the minimal element length and the material parameters shown in Table 1, which is $1.627 \times 10^{-5}$ of the numerical model used in this work. The simulation results shown in Figure 6 confirm that the theoretical reference value is reliable, as the calculated results are almost identical when the increment size is smaller than the critical value. In addition, an increment size greater than the critical value always results in divergence of the calculation. In view of this fact, the increment size was set to be $1 \times 10^{-5}$ in the following section.

*3.2. Model Validation*

For model validation, the calculated nodal displacement of W1~W5 was compared with the test data. A detailed comparison of the node displacement is shown in Figure 7. Some period shifts are observed, which may be related to the deviation between the actual response of the test equipment and the input signal. However, the results indicated that the predicted results agree well with the experimental values in terms of the order of displacement magnitude and variation trend. A more detailed comparison indicates that there are some differences in the nodal displacement. Generally, the predicted value is smaller than the test data, which may be caused by the idealization of the numerical model. As geometry imperfection and manufacturing imperfection were not considered in the numerical model, this may lead to a higher stiffness of the structure. In addition, the structural damping was not considered, even though the influence is marginal, which may still have some minor impact on the energy dissipation. The calculated maximum node displacement occurs at about 10.6 s, and the corresponding deformation and stress distribution are shown in Figure 8. It indicates that the maximum displacement occurs in the

middle overhanging area. Based on the validated results, we consider the numerical model is reliable, which will then be used for extensive parametric studies in the following sections.

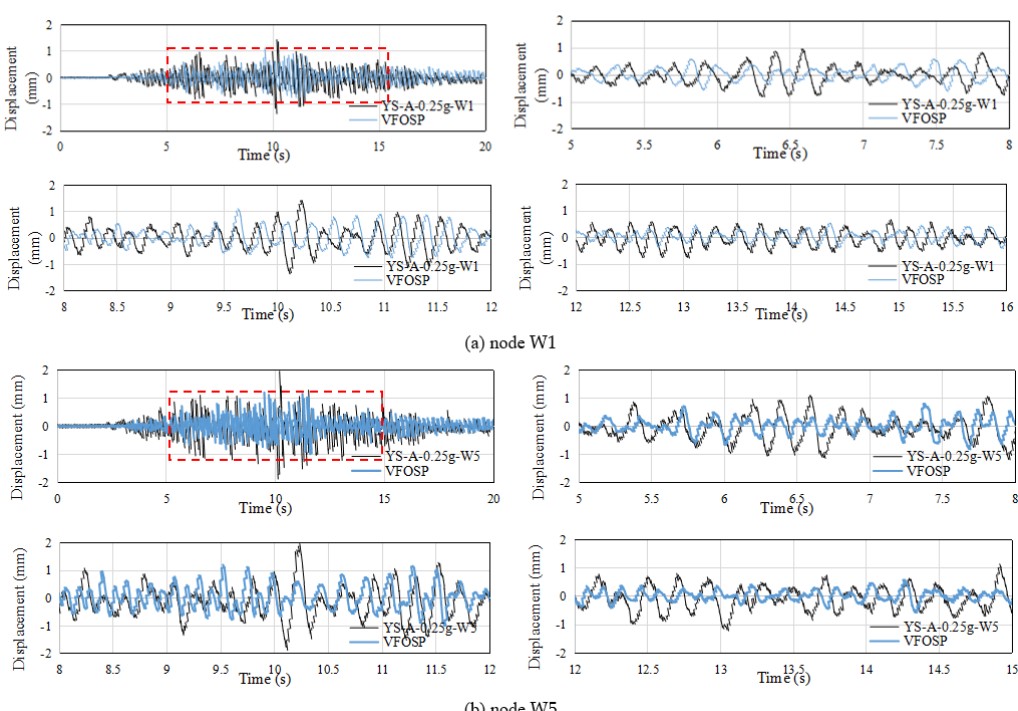

(a) node W1

(b) node W5

**Figure 7.** Comparison of the calculated and monitored node displacement.

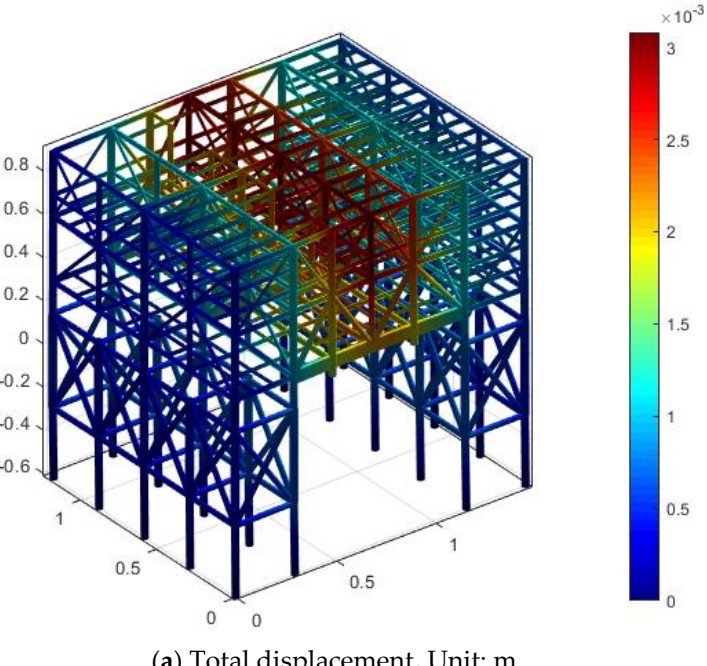

(**a**) Total displacement, Unit: m

**Figure 8.** *Cont.*

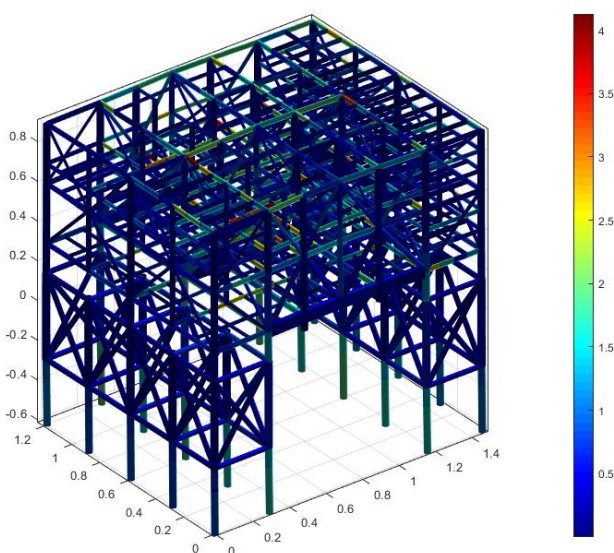

(**b**) Maximum value of the element stress, Unit: MPa

**Figure 8.** Deformed configuration of the OCS ($t$ = 10.6 s, amplification factor = 10).

## 4. Results and Discussion

Based on the numerical method mentioned in Section 2 and the validated code shown in Section 3, systematic parametric studies were carried out to investigate the dynamic characteristics of the OCS under additional loads, which were not covered in the test. The effect of wave damping (water height) and hydrodynamic on the dynamic behavior of OCS were investigated firstly, and then the seismic load and seismic excitation mode were further investigated. Finally, the peak response of the major nodes is shown, which provides detailed information on OCS under typical environmental loads.

### 4.1. Effect of Water Damping and Hydrodynamic

During the service of OCS, the water level varies with high and low tides; in addition, the adhesion of marine organisms will enlarge the hydrodynamic diameter of the pile foundation, and increase the surface roughness. All these factors increase the added water mass and drag force, and may threaten the safety of OCS under extreme load. The natural frequency and damping ratio of OCS will change with the variation in water level [2], which can be obtained by white noise excitation [1,2]. To identify the effect of water damping, the dynamic response of OCS under air and water conditions are compared, and the water depth used is 0.32 m and 0.64 m. Note that the damping parameter is set to be the same for wet analysis ($\alpha$ = 0.908), and the peak seismic load applied is 0.25 g. The simulation results are shown in Figure 9, which indicate that the presence of water accelerates the energy dissipation process, thus reducing the peak displacement. In view of this fact, seismic design based on the atmospheric environment should be safer. The fluid–structure interaction and energy dissipation are most significant when peak displacement occurs. In the case of small acceleration, the hydrodynamic effect is small. Figure 9(a-4,b-4) show the displacement difference between wet and dry, which indicates that the further increase in water depth from 0.32 to 0.64 m has little effect on the displacement response, and the damping coefficient applied, which is smaller than the actual value, may responsible for it. Note that this conclusion is only based on still water, when the wave-induced hydrodynamic is considered; the peak response of OCS may be amplified due to the fluid–structure interaction.

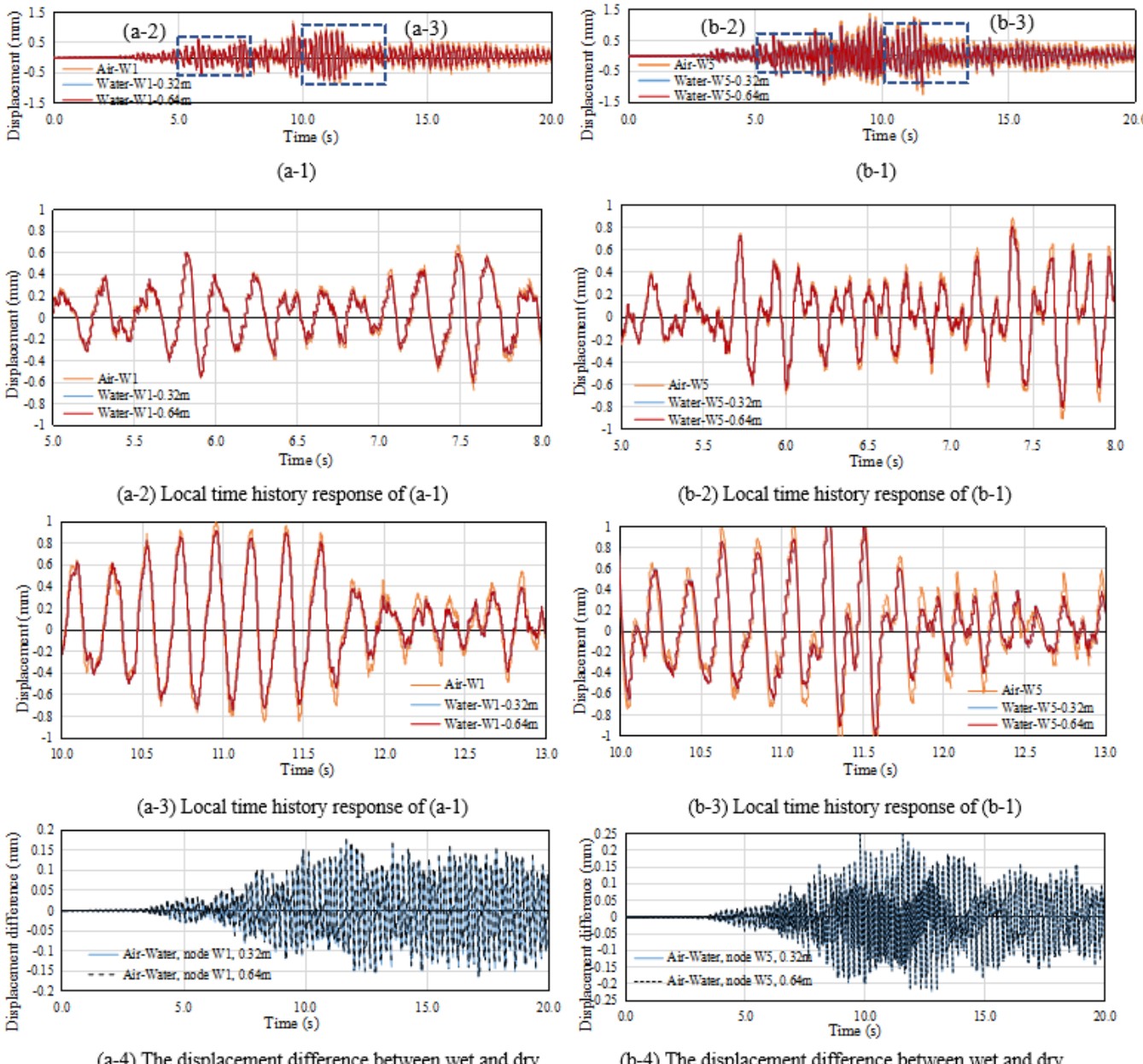

**Figure 9.** Comparison of the displacement history along $X-$axis under dry and wet conditions ($h$ = 0.32 m), the seismic load shown in Figure 5a,b were used.

To figure out the response of OCS under the coupling effect of seismic load and wave load, the effect of wave load was applied. The wave parameters were obtained from the South China Sea [31]; typical values are shown in Table 2. For the scale numerical model, the annual averaged values of the wave parameters were obtained, and then they were scaled accordingly based on the similarity criterion [1,2]. The water depth and seismic load remained the same, and the simulation results are shown in Figure 10. These indicate that the wave load has little impact on the dynamic behavior of OCS when the earthquake intensity is high. Compared to the damping effect of water, the additional load caused by the waves is negligible.

**Table 2.** Wave parameters.

| Parameters | Actual Wave Height, $H$ (m) [31] | Actual Wave Period, $T$ (s) [31] | Simulation Wave Height, $h$ (m) | Simulation Wave Period, $T$ (s) |
|---|---|---|---|---|
| | 3.79~9.65 (Annual averaged $H_s$ = 1.2 m) | 2.1~8.1 (Annual averaged $T$ = 4.0 s) | 0.02 | 0.516 |

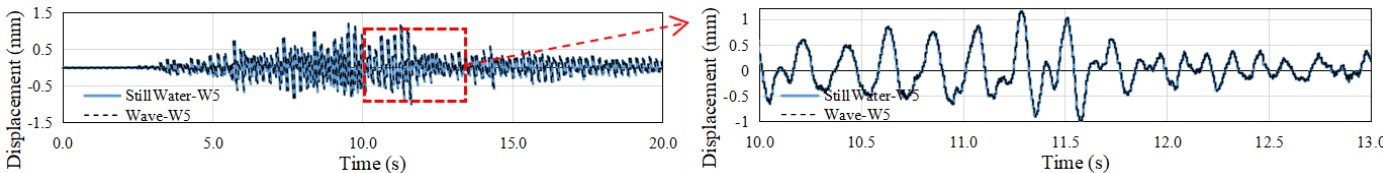

**Figure 10.** Comparison of the displacement history under still water and wave conditions (YS−A−0.25 g, $h$ = 0.32 m, and the wave propagates along the $X$ direction).

To identify the effect of drag force caused by current, a typical current velocity (0.15 m/s, which corresponds to 1.16 m/s for the prototype structure) was applied, and the simulation results are shown in Figure 11. The simulation results indicate that the drag force caused by the current is also negligible compared to the seismic load. Note that the drag force may also be caused by wind load, which was also investigated in this work. However, the simulation results indicate that a steady wind load has little impact on the dynamic behavior of OCS, even though the applied wind velocity is up to 6 m/s for the scale numerical model (corresponding to 46.5 m/s for the prototype structure), in view of this fact, the related results are not shown.

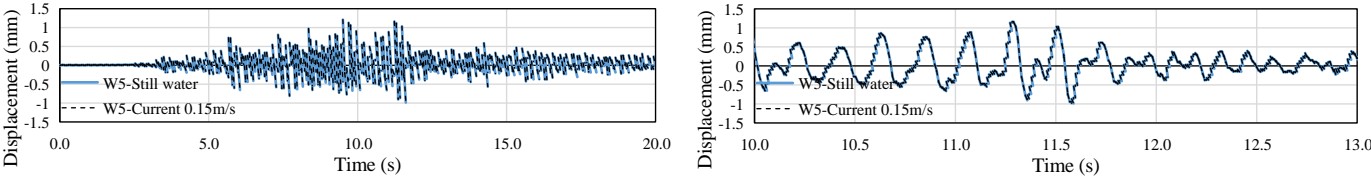

**Figure 11.** Comparison of the displacement history of OCS under the case of without and with current load (YS−A−0.25 g, $h$ = 0.32 m, and the current propagates along the $X$−direction).

### 4.2. Effect of Seismic Load

The earthquake intensity can vary widely in engineering practice, to identify the effect of earthquake intensity on the dynamic behavior of OCS, four intensity levels were investigated. Note that the seismic waves are the same except for their amplitude, as shown in Figure 12. For the seismic direction study, the propagation directions were set along the $X−$ and $Y−$axis, and the peak acceleration of the ground was 0.25 g. The corresponding displacement along the $X−$direction and $Z−$direction is shown in Figure 13 and the corresponding maximum displacement under different earthquake intensities is shown in Figure 14. These indicate that the peak displacement increases almost linearly with the increase in peak acceleration of the ground; this effect is most significant in the $X−$direction. In all cases, the seismic load along the $Z−$axis (vertical direction) results in the downward rigid body displacement of the entire structure; however, the displacement amplitude of the structure around the equilibrium position is much smaller than that along the horizontal direction ($X−$axis).

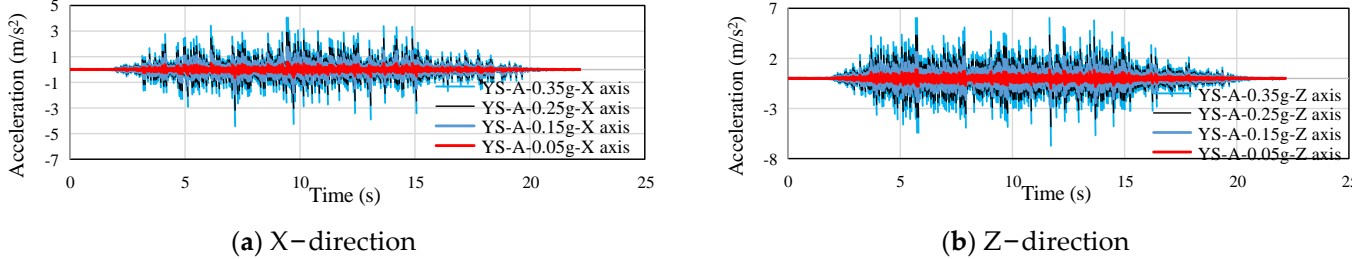

(**a**) X−direction  (**b**) Z−direction

**Figure 12.** Four intensity levels of seismic load used in the simulation generated by the API specification spectrum.

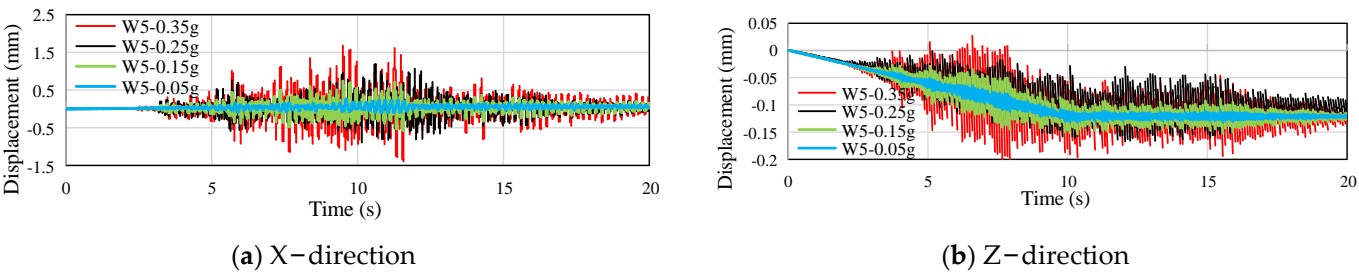

(**a**) X−direction  (**b**) Z−direction

**Figure 13.** Comparison of the displacement history under different seismic intensities.

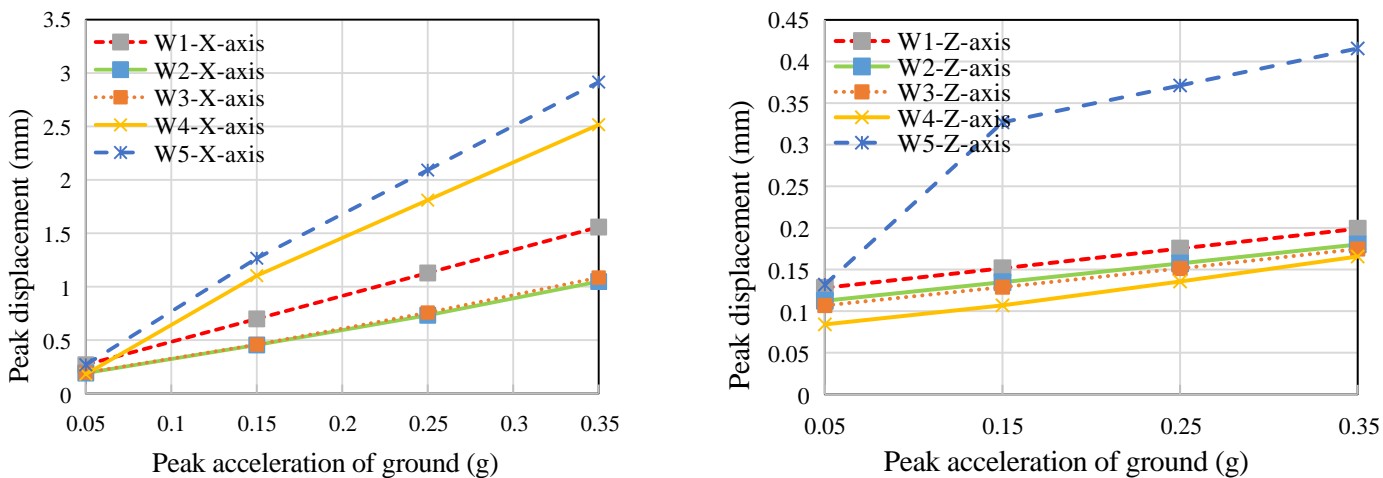

**Figure 14.** Peak displacement of W1~W5 as a function of the peak acceleration of the ground.

Figure 15 shows the time history of the element stress on the tension and compression side. Due to the joint action of axial tension and bending as well as axial compression and bending, the absolute value of stress on the tension side and compression side of the beam element is different. In addition, the time history of stress on the different elements differs a lot; more detailed information on the location of elements can be found in Figure 4. The relationship between the peak stress and the peak acceleration of the ground on the different elements is shown in Figure 16; again, an almost linear relationship can be observed.

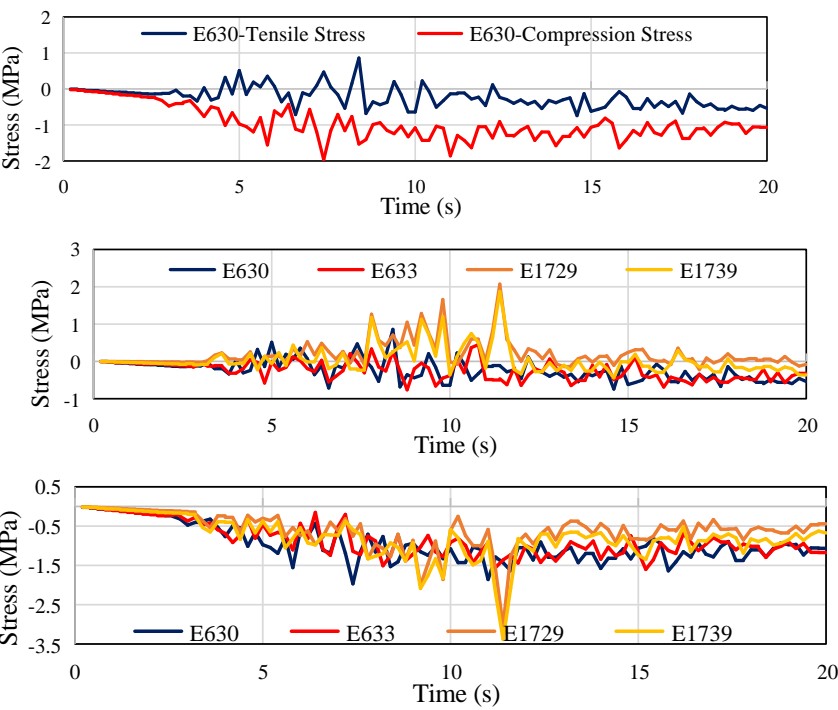

**Figure 15.** Time history of the element stress on the tension and compression side of beam (A positive value indicates tensile stress and a negative value indicates compressive stress).

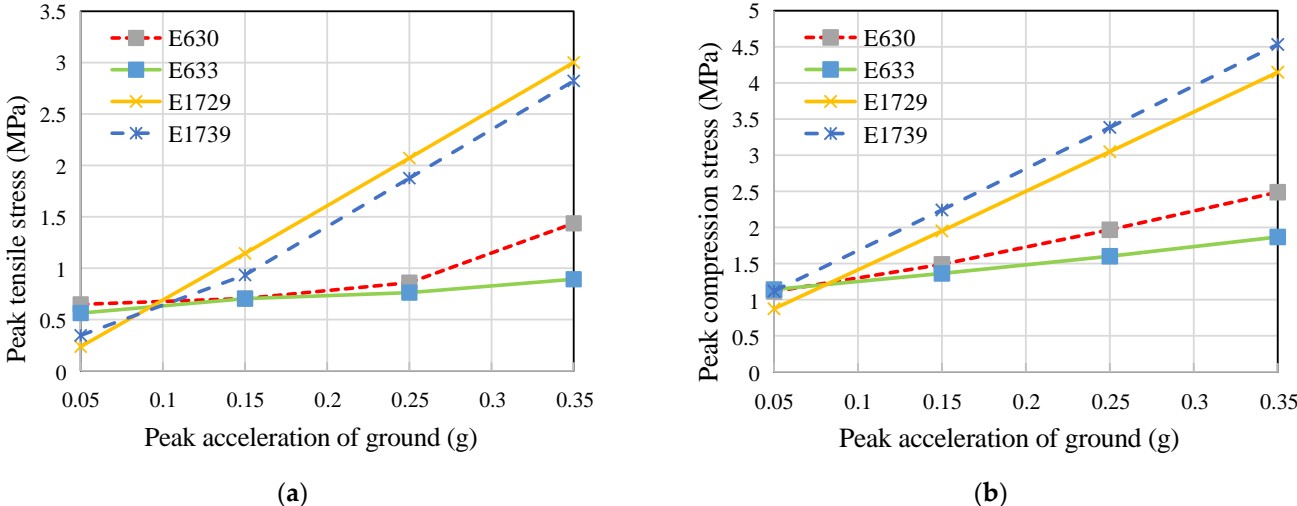

(**a**)　　　　　　　　　　　　　　　　　　　　　　　　　　　　　　(**b**)

**Figure 16.** Peak stress on different elements as a function of the peak acceleration of the ground: (**a**) Tensile stress; (**b**) Compression stress.

### 4.3. Effect of Seismic Excitation Mode

In this part, the effect of the propagation direction of seismic waves on the dynamic behavior of OCS was investigated. A unidirectional seismic load was used, and the seismic waves propagated along the $X-$ and $Y-$axis. The time history of the node displacement is shown in Figures 17 and 18. The simulation results indicate that the node displacements differ a lot when the seismic propagation direction changes. The node displacement along the Y direction is very small under the excitation of the seismic load propagating along the $X-$axis, which is true for all nodes at different heights. However, for a seismic wave propagating along the $Y-$axis, the displacement along the $X-$axis increases with the increase in node height, while the displacement along the $Y-$axis decreases with the increase in node height, which is due to the fact that the structure is not symmetrical with

respect to the YOZ plane. Note that the node displacement is larger when the seismic wave propagates in the $X-$axis, which indicates that the OCS is weaker along the $X-$axis. The time history of the element stress on the tension and compression side under different seismic excitation is shown in Figure 19, which confirms that the seismic propagation direction has a great impact on the structure response.

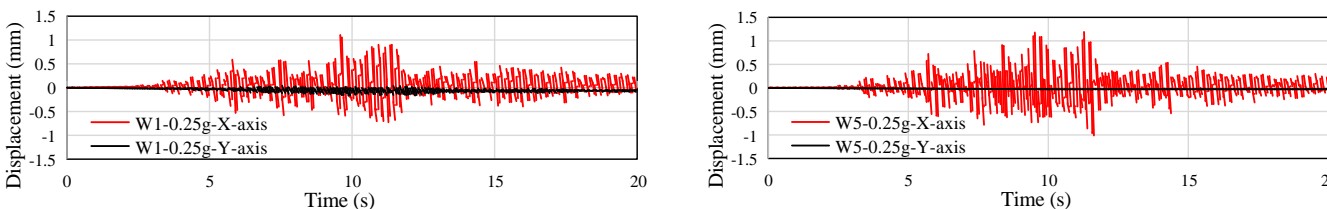

**Figure 17.** Time history of the node displacement under unidirectional seismic load (along $X-$axis).

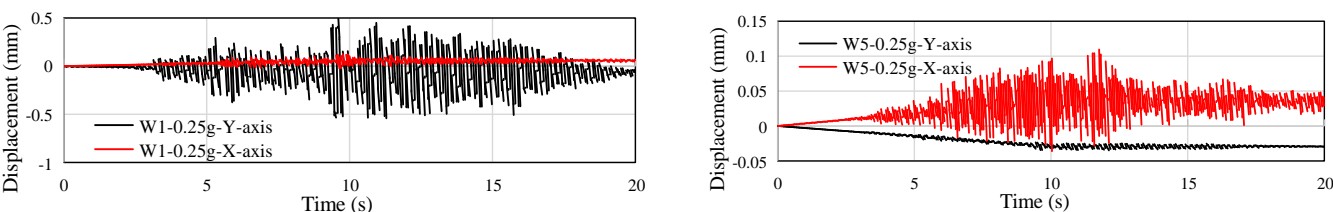

**Figure 18.** Time history of the node displacement under unidirectional seismic load (along $Y-$axis).

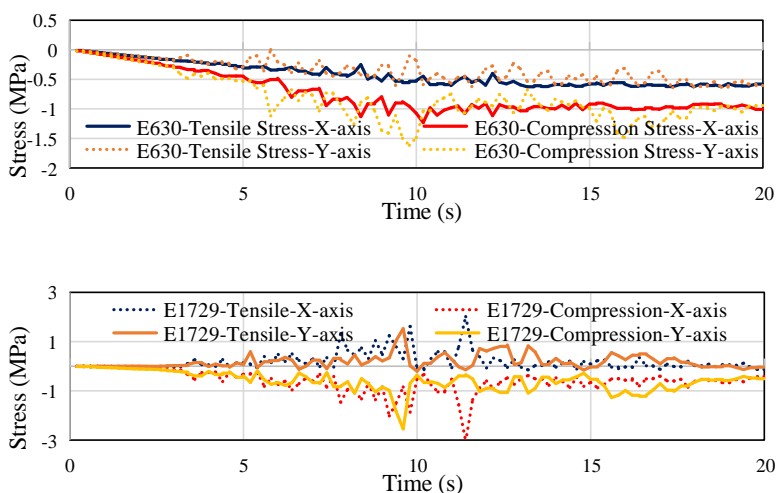

**Figure 19.** Time history of the element stress on the tension and compression side under different seismic excitation.

### 4.4. Peak Response of the Major Nodes

To obtain the overall displacement response of the OCS, and identify the possibility of local structural damage, the response of some key points is monitored. The structures at the intersection of cross sections $X_i$ (i = 0, 2, 4) and $Y_i$, (i = 0, 1, 5, 6) (see Figure 4) are defined as the major structure, and the nodes at the intersection of cross sections $X_i$ (i = 0, 2, 4), $Y_i$, (i = 0, 1, 5, 6) and $Z_i$, (i = 3, 4, 5) are defined as the major nodes. The peak values at these nodes reveal the overall dynamic behavior of the OCS. Multi-valve towers are usually arranged in the large-span valve hall (see Figure 20 dotted box area). The slenderness ratio of the valve tower is large, and brittle ceramic or polymer materials are commonly used for insulation. If the overall stiffness of the support (or suspension) structure is not strong enough, the valve tower may be damaged under strong dynamic action, such as an earthquake [32–34]. In view of this fact, the dynamic response of the valve hall deck is vital

for the safety of the valve tower. As a simplification, the valve tower was not included in the numerical model; instead, a concentrated mass point was applied.

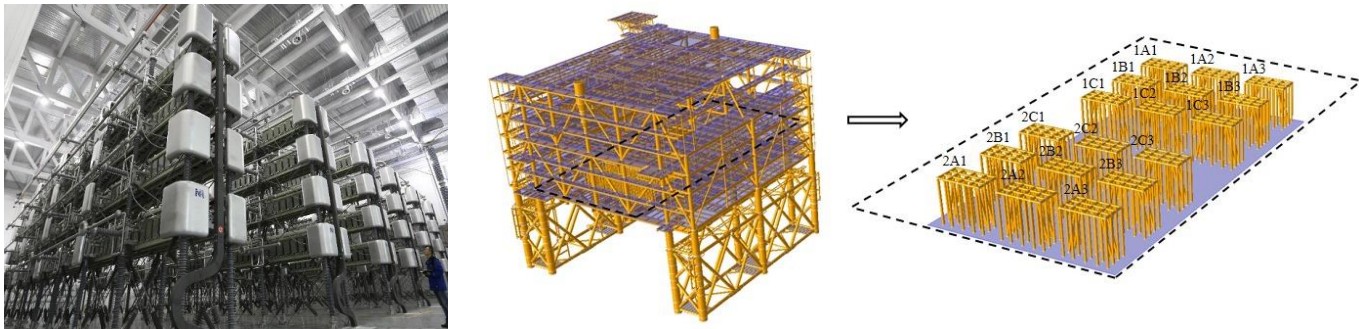

**Figure 20.** Valve tower and the schematic of arrangement.

The peak response of the major nodes is shown in Figure 21; the results indicate that the peak displacement of the different nodes differs a lot even though they are on the same deck. The displacement of the different major nodes under the excitation of peak acceleration is shown in Figure 22. A significant "whipping effect" can be observed in the horizontal direction (marked with a dotted line), which had been observed in previous works [1,2,35], while the "stretching-squeezing" effect can be observed in the vertical direction (also marked with dotted line). For OCS, there are a number of important pieces of electrical equipment mounted on the superstructure. Therefore, reducing the "whipping effect" and "stretching-squeezing" effect of the platform is important for the safety of the system. Generally, the peak displacement increases with the increase in height, and the displacement along the $X-$axis is much higher than that along the $Z-$axis. A more detailed comparison of the displacement and acceleration of some critical nodes is shown in Figures 23 and 24. The results indicate that the displacement and acceleration along the horizontal direction of both nodes are almost the same, while the responses along the vertical direction differ a lot. The displacement and acceleration of node 75 are much higher than that of the major node W2; this difference is caused by the relative lower local rigidity of the large-span valve hall and the inertia force caused by the concentrated mass of the valve tower. In addition, the vibration frequency of vertical response is higher than that of horizontal response. As mentioned earlier, the valve tower is vulnerable to being damaged; in view of this fact, the support structure of the valve hall should be locally strengthened.

*4.5. Discussion*

In this work, the VFIFE method was used for dynamic analysis of an OCS, and different loads were considered. Generally, the predicted displacement magnitude agrees with the test data; however, some period shifts are also observed, which may be related to the deviation between the actual response of the test equipment and the input signal. In addition, the deviation between the test model and the numerical model may also play a role. Note that some simplified assumptions were also made in the numerical model (see Section 2), including pure elastic analysis; smooth surface assumption of the cylinder and the stiffness damping were also ignored. All of these assumptions should be responsible for the deviation between the predicted value and the test data. The pure elastic analysis may be inaccurate under large seismic loads, as local plastic deformation may occur. In addition, the hysteretic energy dissipation characteristics under earthquakes should be further considered. Moreover, only the scale model is used in the current study. In our future work, the VFIFE method will be used to analyze the dynamic behavior of the prototype structure, and more model details, such as elastic-plastic analysis and structure–soil coupling analysis, will be further considered.

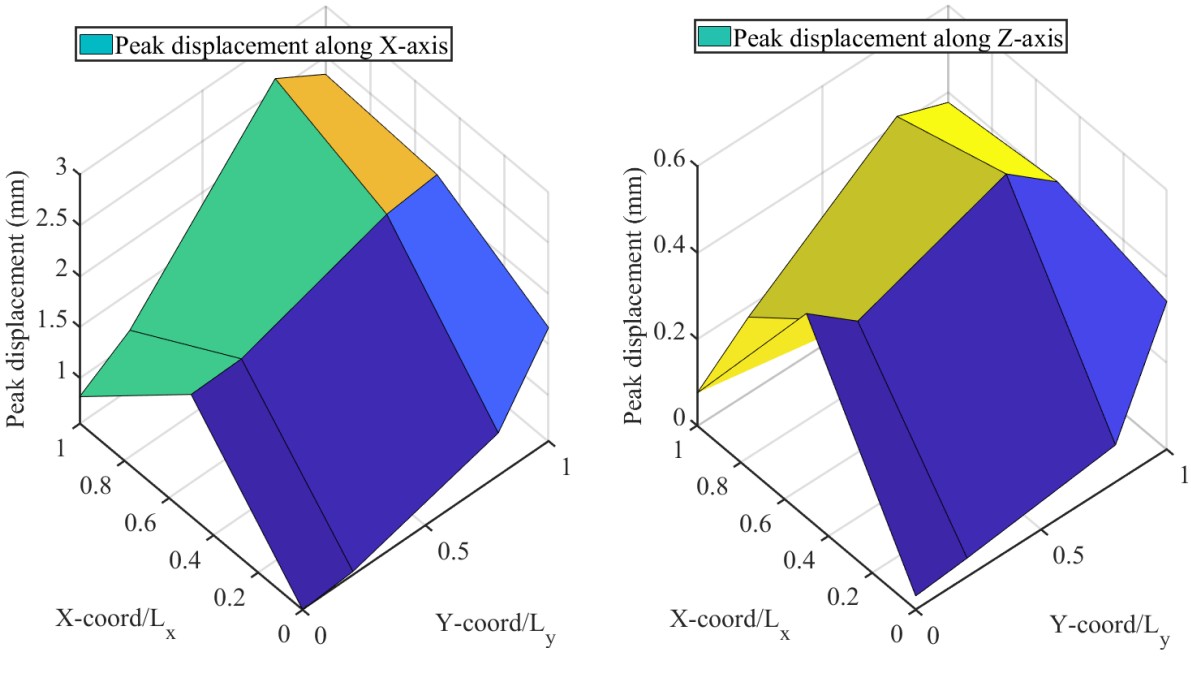

(**a**) Height $Z_3$

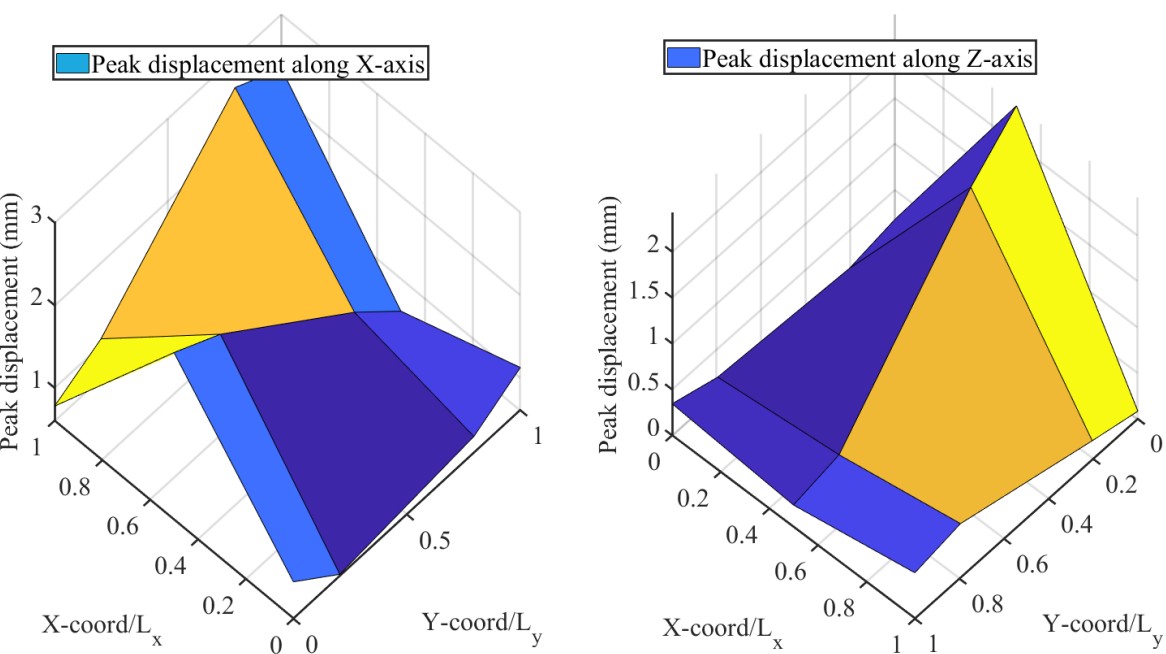

(**b**) Height $Z_4$

**Figure 21.** *Cont.*

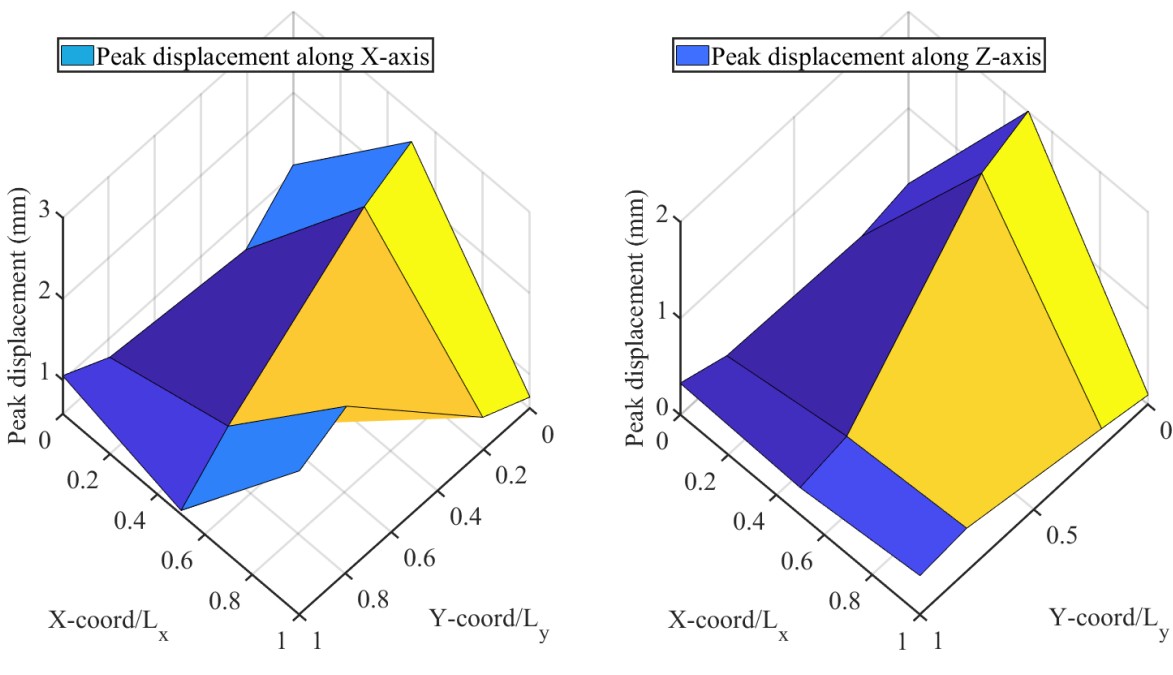

(**c**) Height $Z_5$

**Figure 21.** Comparison of the peak response of the major nodes on different decks (YS−A−0.25 g, $h = 0.32$ m, Note: the time corresponding to the peak displacement is not necessarily the same, the $X-$coord$/L_x$ and $Y-$coord$/L_y$ represent normalized values along the length and width directions of OCS).

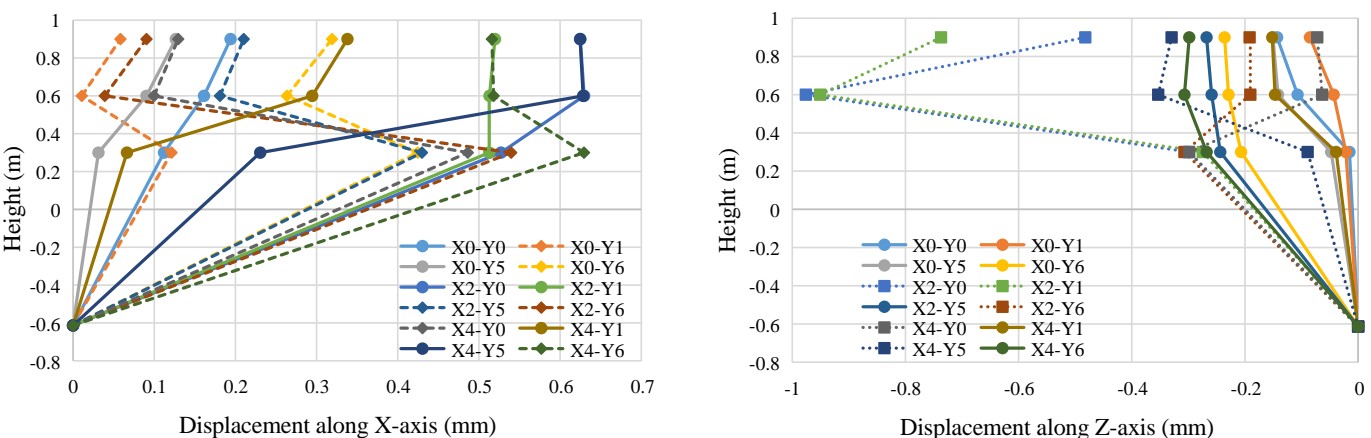

**Figure 22.** The node displacement along the height direction of different piles at peak acceleration.

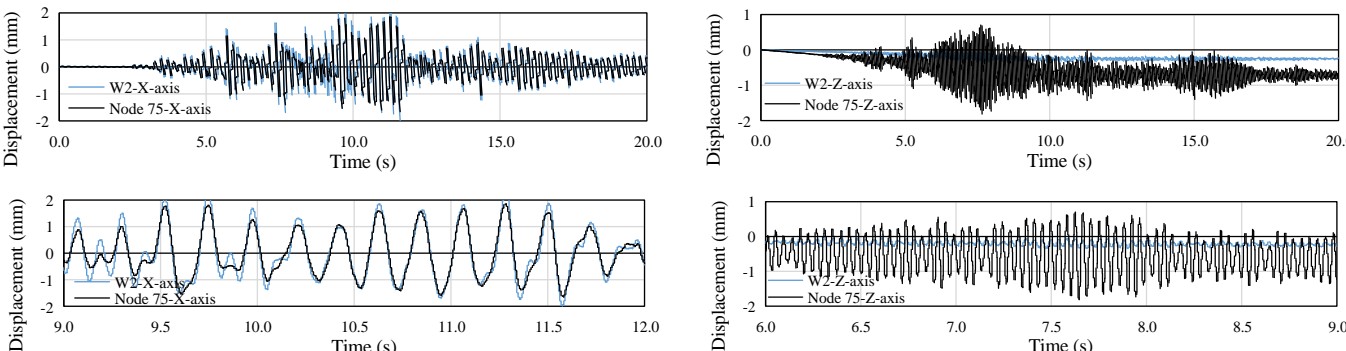

**Figure 23.** Comparison of the displacement response of major nodes and the node on the deck of the valve hall (Node 75).

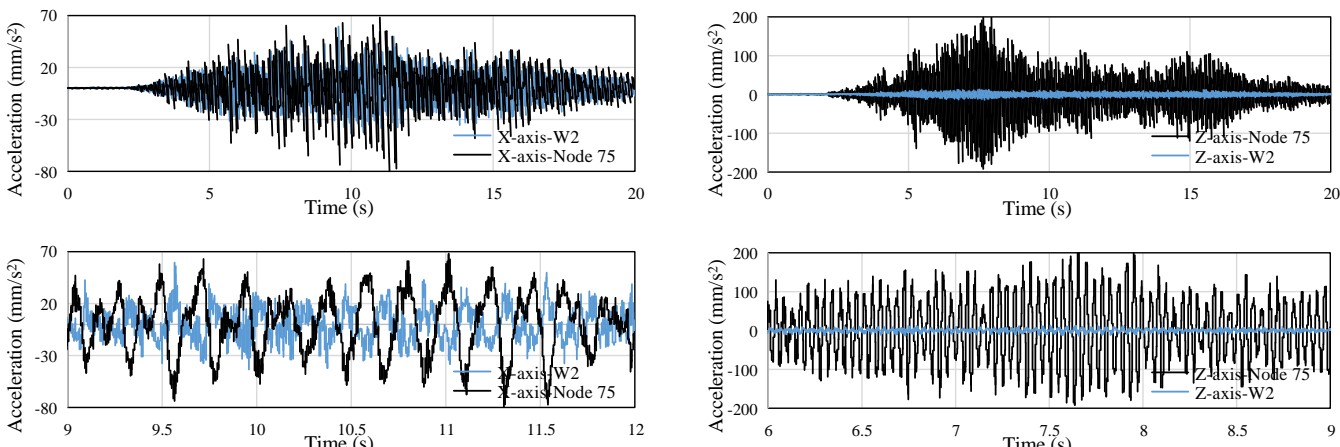

**Figure 24.** Comparison of the acceleration response of major nodes and the node at the valve hall (Node 75).

## 5. Conclusions

In this study, a user-defined in-house FORTRAN code based on the Vector Form Intrinsic Finite Element (VFIFE) method was developed, which can be used for dynamic analysis of offshore structures that experience earthquake, wind and hydrodynamics loads. After model validation, the dynamic behavior of OCS under different loading conditions was carefully studied, based on the present study, the following conclusions can be drawn:

(1) The proposed numerical simulation method is reliable.
(2) The presence of water accelerates the energy dissipation process, thus reducing the peak displacement; however, the wave load and the drag force caused by current and wind have little effect on the dynamic behavior of OCS when the seismic load is high.
(3) The peak displacement and stress increase almost linearly with the increase in peak acceleration of the ground; and the seismic direction has a great impact on the dynamic behavior of the OCS, which is caused by the asymmetry of the structure.
(4) Both a "whipping effect" and "stretching-squeezing effect" were observed during the earthquake, and the vertical acceleration response of the valve hall deck was much higher than other structures; local reinforcement should be made to protect the electrical equipment.

Note that the current version of the proposed in-house FORTRAN code can also be used for elastoplastic and structure–soil interaction analysis. As this work mainly focuses on elastic small deformation analysis, the elastoplastic analysis was not conducted. The biggest advantage that the user-defined in-house FORTRAN code has is that it is suitable for large deformation analysis (as there are frequent typhoons along the southeast coast

of China, collapse analysis of the platform is one of the application scenarios) and has robust convergence (this is the inherent advantage of the VFIFE method compared to the traditional finite element method).

**Author Contributions:** Conceptualization, Z.S., Y.Y. and H.W.; Methodology, Z.S. and Y.Y.; software, H.W.; Validation, H.W., S.H. and J.C.; Formal analysis, Z.S., Y.Y. and H.W.; Writing—original draft preparation, Z.S. and H.W.; Writing—review and editing, Z.S., Y.Y. and H.W.; Funding acquisition, Z.S. All authors have read and agreed to the published version of the manuscript.

**Funding:** This project was supported by the Natural Science Foundation of Zhejiang Province (No. LQ21E090010), the Open Fund of Key Laboratory of Far-shore Wind Power Technology of Zhejiang Province (ZOE2020002), Natural Science Foundation of Fujian Province (2021J05004,2020J01010), President's Foundation of Xiamen University (20720210068). This research received no external funding.

**Institutional Review Board Statement:** Not applicable.

**Informed Consent Statement:** Not applicable.

**Data Availability Statement:** There is no additional data supporting reported results.

**Conflicts of Interest:** The authors declare that they have no known competing financial interests or personal relationships that could have appeared to influence the work reported in this paper.

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
