# Peer review of "Dynamic Response Analysis of Offshore Converter Station Based on Vector Form Intrinsic Finite Element (VFIFE) Method"

_jmse, doi:10.3390/jmse10060749_

Round 1

Reviewer 1 Report

The reviewer wants to thank the author for the paper presenting a numerical investigation of different structural loads on an offshore convertor station:

*1) The citation style needs adaptation hence JSME requires a numbering of the literature references based on the occurrence in the text and not alphabetical. Please check the journal’s guidelines.

*2) Line (L) 13: The authors use “user defined program” and the reviewer is not sure, if this is the correct name for it. The software is fully written by the authors using FORTRAN as the programming language, correct? Maybe bespoke software based on FORTRAN? Maybe check comparable literature.

*3) L29: wind power is not a power transmission but a power generation and requires specific transmission.

*4) Table 1: please integrate this table in one sentence in the text.

*5) L63: please specify a tsunami wave with 45 degrees? What does the angle describe?

*6) Starting with L64: This sentence includes 16 literature references. Please clarify why each one is needed.

*7) L83: This paragraph is very important hence it clarify the connection to previous work as well as the novelty of this paper. Nevertheless, the explanation falls short. Both experimental and numerical analysis were done previously and the novelty is now the new FORTRAN code? Please clarify this hence this is important to understand the novelty of this paper.

*8) L116: Please add the references for the stiffness damping.

*9) The first part of section 3 had some problems with the formatting (reviewer’s pdf). Please check and correct this.

*10) Please include the water depth in Figure 4 as well as a full introduction of the local coordinate system.

*11) Why is there only one water depth in Table 2 and L283 mentions two?

*12) Does the reviewer understand this correctly: The authors validate their code with experimental investigations of an earthquake and based on this investigated further additional loads, which were not covered in the experiment?

*13) The section 4 should start with an overview of the following results, which provides the connection between each subsection.

*14) Figure 9: Why are there different time windows used for 2 and 3? Could you please also provided the difference between wet and dry? At least statistically with the min and max values. Now the comparison is really hard to see. Furthermore, it would make more sense to combine wet and dry condition for one case in a figure to allow a direct comparison than group them together.

*15) Figure 10: the two cases seem to be identical? How are the seismic loads and the waves combined? How is ensured that the worst case is reached?

*16) L316: the reference Fig.1 is clearly incorrect and there seems to be an issue with the following Figure.

*17) Section 4.2 and following: it is not clear why wave, water depth, wind speed and so on is used for the specific investigations. Please go through the paper and make this clear.

*18) General: The image quality of many figures and graphs is very low and should be improved. Please check at least for covered legends.

*19) Please include a dissuasion section to reflect on the methodological approach.

*20) Again in the conclusion the specific novelty of this paper should be further highlighted.

*21) Please explain the advantage of the FORTRAN code in relation to other commercial applications, which are capable to simulate such a structure. Is it intended to publish the full code? Please include a specific data statement.

Reviewer 2 Report

It is very difficult to evaluate the paper on technical basis because it needs a huge revision on writing and style. Authors should take care with the paper presentation - it seems that no one revised the paper before submitting. Avoid long sentences, several words were written wrongly, some sentences are incomplete.

Some other important points that need to be improved:

  • abstract must give an idea of what kind of analysis was performed.
  • model should be better described.
  • Some figures must be updated. 

Reviewer 3 Report

  1. Some figures (Figures 1, 4 and 8) are unclear and should be redrawn.
  2. How do authors decide the values of added mass and drag coefficients shown in Eq. 9 and Eq.11? Please explain it.
  3. Could authors explain the direction of and in Eqs. 7 and 8?
  4. The horizontal wave forces on the jacked platform are displayed in Eq. 9. How about wave forces and in y- and z-direction? Besides, the rods on frame model are not vertical, how to decide the values of added mass and drag coefficients?
  5. I don’t know whether it is significant or not, as the acceleration of seismic wave reach 0.25g and water depth is assume as 0.32m and 0.64m .Please explain it.

Round 2

Reviewer 1 Report

The reviewer thanks the authors for their corrections and answers. The quality of the paper could be significantly increased. S/he understands the advantages of a FORTAN code in relation to a commercial code in general, but s/he hoped that the authors could be more specif. Is there anything new in the FORTRAN code or is it only a replication? Why should anybody be excited about this when it only replicates a well-used and tested commercial code? Will it be publicly available? Thank you!

Author Response

Dear reviewer, it is a very good question.

Note that this project is supported by the Key Laboratory of Far-shore Wind Power Technology of Zhejiang Province, aiming at developing a proprietary software with independent intellectual property rights for offshore wind power structure analysis.

The current version of this software can also be used for elastoplastic analysis and structure-soil interaction analysis (soil spring model), as this work mainly focus on elastic small deformation analysis, the elastoplastic analysis was not conducted. According to the authors’ opinion, the most advantage of the user defined FORTRAN code is that it suitable for large deformation analysis (as there are frequent typhoons along the southeast coast of China, collapse analysis of the platform is one of the application scenarios) and has strong convergence (This is the inherent advantage of vector finite element method compared with traditional finite element method), besides, as far as the author knows, few proprietary software based on vector finite element method is available.

One should also notice that, the functionality of this software is still being improved, and more accurate models will be added in future releases, and more complex problems can be simulated in the coming version.

Once the accuracy of the code is fully verified, we intend to release a version in the future.

Reviewer 3 Report

All questions are well answered. No further comments.

Author Response

Dear reviewer, thank you for your comments.